# ATTENTION INCORPORATE NETWORK: A NETWORK CAN ADAPT VARIOUS DATA SIZE

## ABSTRACT

In traditional neural networks for image processing, the inputs of the neural networks should be the same size such as 224×224×3. But how can we train the neural net model with different input size? A common way to do is image deformation which accompany a problem of information loss (e.g. image crop or wrap). In this paper we propose a new network structure called Attention Incorporate Network(AIN). It solve the problem of different size of input images and extract the key features of the inputs by attention mechanism, pay different attention depends on the importance of the features not rely on the data size. Experimentally, AIN achieve a higher accuracy, better convergence comparing to the same size of other network structure.

## 1 INTRODUCTION

Human use the attention mechanism to recognize the world, pay different attention on different region of a image. The mixture of attention and neural network has been studied in the previous literature (Zhao et al., 2017; Itti & Koch, 2001; Shin et al., 2016; Wang et al., 2017). Attention-based network is widely used in sequence model to process text or audio (Mnih et al., 2014).

In this paper we propose Attention Incorporate Network(AIN), AIN have a huge advantage in image classification problem especially for different size of input. Recent advances of image classification focus on a "very deep" network structure, from AlexNet (Krizhevsky et al., 2012) to DenseNet (Huang et al., 2016) the precision become higher and higher, but some problems are still remain to be solved, our model exhibits following appealing properties to solve these problem:

**(1)**Many people comparing Neural Network to human neurons, human can recognize the object easily no matter what the image shape is. But all kinds of popular network like AlexNet (Krizhevsky et al., 2012),VGG (Simonyan & Zisserman, 2014), Inception (Szegedy et al., 2016)and ResNet (He et al., 2016) couldn't process the image in different size. An engineering solution is to scale the image into a same shape, which isn't an aesthetic way due to human neurons won't scale the object image while recognizing it. how does our network structure AIN training feed-forward is that: combine the attention matrix and image together and give a label after incorporate all the key information(Fig. 1). The input image can be arbitrary size due to the AIN feed-forward operation which gives a more natural solution than traditional image scaling(Fig. 1).

**(2)**In modern convolutional network structure, image size will decrease while convolution layer get deeper. For example, the input image size of AlexNet (Krizhevsky et al., 2012) is 224×224×3 and the output of the last convolutional layer is 7×7×4096. The spatial information of image will be transferred into the different channels. Pooling and Convolution with stride are the well known layers to reduces the image size. AlexNet (Krizhevsky et al., 2012) used maxpooling, Resnet (He et al., 2016) used Conv1×1 with stride of 2. Information will remain only $\frac{1}{s^2}$ (s=stride or pooling size) during feed-forward the transition layer. In AIN we used *attention incorporate layer*(AIL) to extract the key features of the image without information loss. Experimentally, AIN achieved a lower error comparing to pooling or covolution with stride in the same network structure.

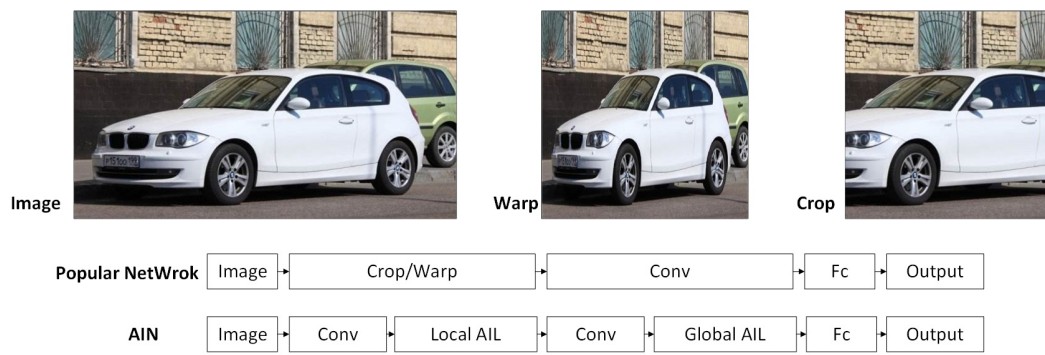

Figure 1: Top:croping or wraping into a fixed size. Bottom:popular network and AIN structure.

## 2 RELATED WORK

The field of attention is one of the oldest in psychology. Evidence from human perception process(Mnih et al., 2014) show the important of the attention mechanism which uses top information to guide bottom-up feed-forward process(Wang et al., 2017). Attention was widely used in many applications: Recommendation system (Vinh et al., 2018), Image classification (Asada, 2014; Zhao et al., 2017), Image caption generation (Xu et al., 2015), Machine translation (Luong et al., 2015; Firat et al., 2016), Speech recognition (Chorowski et al., 2015; Bahdanau et al., 2016) and gave us some good result.

In Image classification, people tried some method to solve the arbitrary input size. He, K. et. al (He et al., 2015). used *spatial pyramid pooling*(SPP) to remove the fixed-size constraint of the network. Pooling size and stride depends on the input data to get a fixed output at the last convolutional layer. The main idea of their network is artificially divide image into fixed numbers of region. The disadvantage of SPP is that pooling layer lost lot of information while training feed-forward especially for those *"very deep"* networks.

Pooling reduces the size of the hidden layers so quickly, stacks of back-to-back convolutional layers are needed to build really deep networks. Zeiler et. al (Zeiler & Fergus, 2013) introduce Stochastic Pooling where the act of picking the maximum value in each pooling region is replaced by a form of size-biased sampling. Graham et. al (Graham, 2014) introduce fractional max-pooling (FMP). The idea of FMP is to reduce the spatial size of the image by a factor of $\alpha$ with $1 \leq \alpha \leq 2$. Their solutions reduce the information loss of the pooling layer but still left the problem behind.

## 3 ATTENTION INCORPORATE NETWORK

Our Attention Incorporate Network constructed by stacking multiple *local attention incorporate layer*(LAIL) and a *global attention incorporate layer*(GAIL) before the FC-layer. The output of LAIN can be any size depend on the input, and the output of GAIN will be the fixed size design by the network structure to ensure that no size conflict between the GAIN and FC-layer. And for both GAIL and LAIL share exactly the same mathematical calculation.

### 3.1 ATTENTION INCORPORATE LAYER

**Forward Pass.** Traditional convolutional feed-forward networks connect the output of the $\ell^{th}$ layer as input to the $(\ell + 1)^{th}$ layer (Krizhevsky et al., 2012). The transition layer in AlexNet (Krizhevsky et al., 2012), Resnet (He et al., 2016) fellow the equation(1)(2):

$$X_{\ell+1} = MaxPooling(X_\ell) \tag{1}$$

$$X_{\ell+1} = Cov(X_\ell, stride = 2) \tag{2}$$

We propose AIL instead of those traditional sharpen transition layer. Suppose that $X_{in}$ is the input of AIL with the size of ($M$,$N$,$c$), $X_{out}$ is the out of AIL with the size of ($\frac{M}{s}$,$\frac{N}{s}$,$c$'), and the mathematical

principle of AIL follow these equations:

$$X = Relu(Cov(X_{in}, 1 \times 1)) \tag{3}$$

$$W = Sigmoid(SeparableConv(X_{in}, k \times k)) \tag{4}$$

X in equation(3) extract the content matrix of the input with the operation RELU after Cov1×1. W in equation(4) extract the attention matrix with the operation Sigmoid after SeparableConv k×k. Both of them have the size of (*M,N,c'*).

The motion of choosing this setting is that the interaction between channel is enough for transmit the information in content matrix X. We design the W matrix as the mixture of spatial information, **k** in equation(4) refer to the kernel size which is a hyper-parameter in AIN and it should be an integer greater than 1. Experimentally we choose **k=3** as the default value by fine tuning k in our three experiments. Separable Convolution which made up of Depthwise (Chollet, 2016) and Conv1×1. We choose it for the reason of model compression.

Window shifting is one of the basic operation in convolutional networks, the operation is shown in Fig. 2. For mathematically intuitive, we are going to do the calculation on one of the selected window. Suppose that $\tilde{X},\tilde{W}$ is one of the selected window from X,W.

$$\tilde{X}, \tilde{W} = Window(kernel\_size = (m, n, c'), stride = s)(X, W) \tag{5}$$

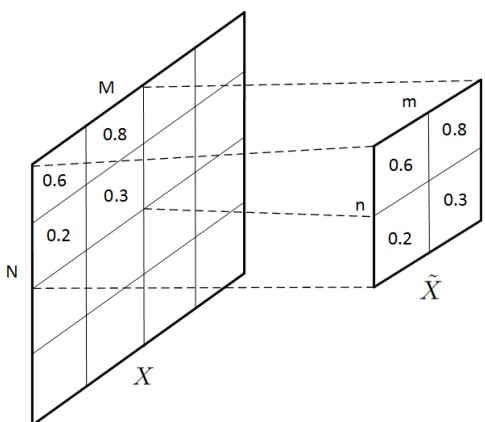

Figure 2: Select Window Operation.

Both of $\tilde{X},\tilde{W}$ have the size of (*m,n,c'*). Suppose that X$_{out}$ is the output of AIL with the shape of (1,1,*c'*).

$$\tilde{X}_{out,k} = \frac{\sum_i^m \sum_j^n (\tilde{W}_{i,j,k} \cdot \tilde{X}_{i,j,k})}{\sum_i^m \sum_j^n \tilde{W}_{i,j,k} + \epsilon} \tag{6}$$

$\tilde{X}_{out,k}$ stands for the value of $\tilde{X}_{out}$ in k$^{th}$ channel. $X_{out}$ equals to the stack by $\tilde{X}_{out}$ from all the sampling window in equation(5). $\epsilon$ is a small number equals to $10^{-8}$ to make sure the fraction won't divide by zero. Kernel size *m,n* can arbitrary value during the feed-forward calculation. If the kernel *m=n=*constant then the calculation present as Conv+Pooling with attention mechanism, we name it as *local attention incorporate layer*(LAIL). If *m=M* and *n=N*, there is only one window produced during the sampling, the window represent the whole image, the calculation extract the global information of the input image. We name it as *global attention incorporate layer*(GAIL). For more concretely, the forward operation shows the following(Fig. 3)

**Backward Pass.** $\tilde{X},\tilde{W}$ in equation(5) have different representation of content and weight. These matrix learnt by the backward equation(7)(8):

$$\frac{\partial \tilde{X}_{out,k}}{\partial \tilde{X}_{x,y,k}} = \frac{\tilde{W}_{x,y,k}}{\sum_i^m \sum_j^n \tilde{W}_{i,j,k} + \epsilon} \tag{7}$$

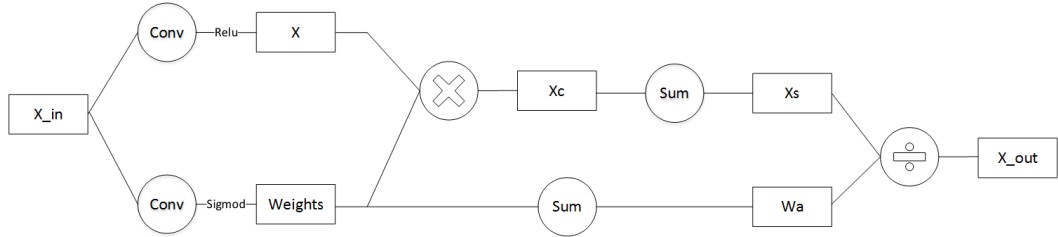

Figure 3: AIL computational graph.

$$\frac{\partial \tilde{X}_{out,k}}{\partial \tilde{W}_{x,y,k}} = \frac{\sum_i^m \sum_j^n \tilde{W}_{i,j,k} \cdot (\tilde{X}_{x,y,k} - \tilde{X}_{i,j,k})}{(\sum_i^m \sum_j^n \tilde{W}_{i,j,k}^2 + \epsilon)} \tag{8}$$

$\frac{\partial \tilde{X}_{out,k}}{\partial \tilde{X}_{x,y,k}}$ stands for the gradient in the $\tilde{X}_{x,y,k}$ passed by $\tilde{X}_{out,k}$, it's pretty obvious that if the weight matrix W has a high value in point(x,y,k), the gradient of content X in this point will also have a high value. In global vision that if there are something important in this image, AIL will give a high gradient on those important area.

$\frac{\partial \tilde{X}_{out,k}}{\partial \tilde{W}_{x,y,k}}$ stands for the gradient in the $\tilde{W}_{x,y,k}$ passed by $\tilde{X}_{out,k}$, from the equation(8) we can know that AIL will likely to give a higher gradient to the point(x,y,k) if constant $\tilde{X}$ in this point has a high variance to other points.

## 3.2 NETWORK STRUCTURE

DenseNet (Huang et al., 2016) achieve a high accuracy in image classification problem. We use the *Dense Block* as the basic block in AIN for the reason that DenseNet is widely used in image classification problem and require substantially fewer parameters and less computation to achieve a reasonable good performances. We implement AIN in the kaggle competition of *Furniture-128*[1] which has various image size for training. The specific networks are shown in Table. 1

Table 1: AIN architectures for kaggle competition of *Furniture-128*

| layers | Output Size | AIN-121 | AIN-169 |
|---|---|---|---|
| Input | M×N×3 | - | |
| Cov | $\frac{M}{2} \times \frac{N}{2} \times 64$ | kernel_size7×7,stride2 | |
| LAIL(1) | $\frac{M}{4} \times \frac{N}{4} \times 64$ | kernel_size3×3,stride2 | |
| Dense Block(1) | $\frac{M}{4} \times \frac{N}{4} \times 256$ | $\binom{1\times1cov}{3\times3cov} \times 6$ | $\binom{1\times1cov}{3\times3cov} \times 6$ |
| LAIL(2) | $\frac{M}{8} \times \frac{N}{8} \times 64$ | kernel_size3×3,stride2 | |
| Dense Block(2) | $\frac{M}{8} \times \frac{N}{8} \times 256$ | $\binom{1\times cov}{3\times3cov} \times 12$ | $\binom{1\times1cov}{3\times3cov} \times 12$ |
| LAIL(3) | $\frac{M}{16} \times \frac{N}{16} \times 128$ | kernel_size3×3,stride2 | |
| Dense Block(3) | $\frac{M}{16} \times \frac{N}{16} \times 512$ | $\binom{1\times1cov}{3\times3cov} \times 24$ | $\binom{1\times1cov}{3\times3cov} \times 32$ |
| LAIL(4) | $\frac{M}{32} \times \frac{N}{32} \times 256$ | kernel_size3×3,stride2 | |
| Dense Block(4) | $\frac{M}{32} \times \frac{N}{32} \times 1024$ | $\binom{1\times1cov}{3\times3cov} \times 16$ | $\binom{1\times1cov}{3\times3cov} \times 32$ |
| GAIL | $1 \times 1 \times 512$ | kernel_size$\frac{M}{32} \times \frac{N}{32}$ | |
| FC,Softmax | 128 | | |

[1]https://www.kaggle.com/c/imaterialist-challenge-furniture-2018

**Local Attention Incorporate Layer.** The size of input tensors will decrease by $\frac{1}{s}$ (s=stride) through LAIL, *matrix Weights* produce by Cov+Sigmoid which gives a value between 0-1 as an attention gate, *matrix X* produce by standard Cov+Relu which contains the key information of the image. The weighted average output is the sum of the *matrix X* each weighted by *matrix Weights*. A simple visualization example of LAIL is shown in Fig. 4.

**Global Attention Incorporate Layer.** GAIL was designed to solve the arbitrary inputs problem. The output of GAIL will be a fixed size such as $1 \times 1 \times 512$ what ever the input size is. The computational logic of GAIL and LAIL are exactly the same. GAIL is the last layer of convolutional layers, the feed-forward feature will be highly abstract (e.g. Whether it is a object or the unimportant background). The image information will be transferred to the fixed number of channels to make sure that where will be no conflict between GAIL and Fully-Connected Layer. A visualization example of GAIL is shown in Fig. 5.

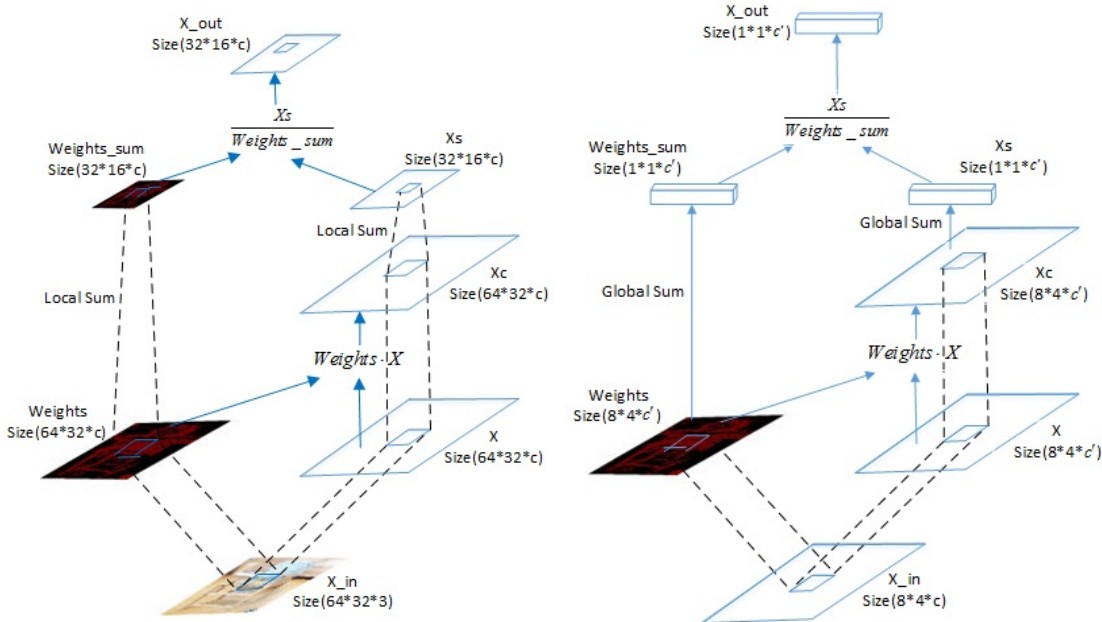

Figure 4: LAIL computational graph.  Figure 5: GAIL computational graph.

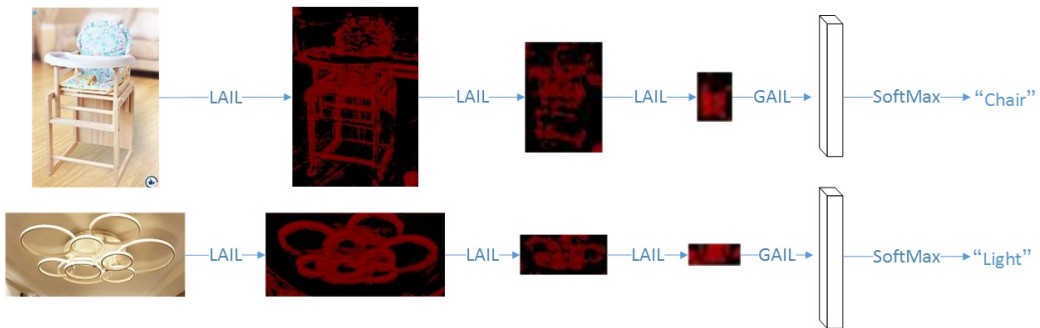

Figure 6: Visualization of AIN.

**Implement Details.** All image data set can use the network structure in Table. 1 by tuning some model parameters (e.g. number of AIL, number of class). Experimentally we used AIN-121 to train the model in kaggle competition of *Furniture-128*. Image size in the data set have various size. AIN-121 include 4×LAIL, 4×DenseBlock and 1×GAIL before the Fully-Connected layer. All of

the LAIL have the kernel_size of 3×3. Dense Block are exactly the same as DenseNet (Huang et al., 2016). Part of the training data visualized in Fig. 6

# 4 EXPERIMENTS

In this section we implement AIN in three datasets: CIFAR-10, Kaggle-*Furniture128*, ImageNet. Given the limited computational resource and parameters, we compared the performance of AIN to baseline architectures ResNet and DenseNet.

**Data argument** Data argument is a widely used technique to get a better performance in image classification problem. There are various image sizes in ImageNet and Kaggle-*Furniture128*, data augmentation scheme for training images introduced in (He et al., 2016; Huang et al., 2016) include: scale augmentation, random crop, horizontal flip. Scale/crop fixed the various image size problem. We denote this data augmentation scheme by a + mark at the end of the dataset name (e.g. ImageNet+). To take advantage of AIN ,we introduce **zoom** for data argumentation and make a comparison to scale/crop . The illustration of crop and zoom shows in Fig. 7. Other image pre-processing method will stay exactly the same as (He et al., 2016; Huang et al., 2016).We denote this new data augmentation scheme by a ∗ mark at the end of the dataset name (e.g. ImageNet*)

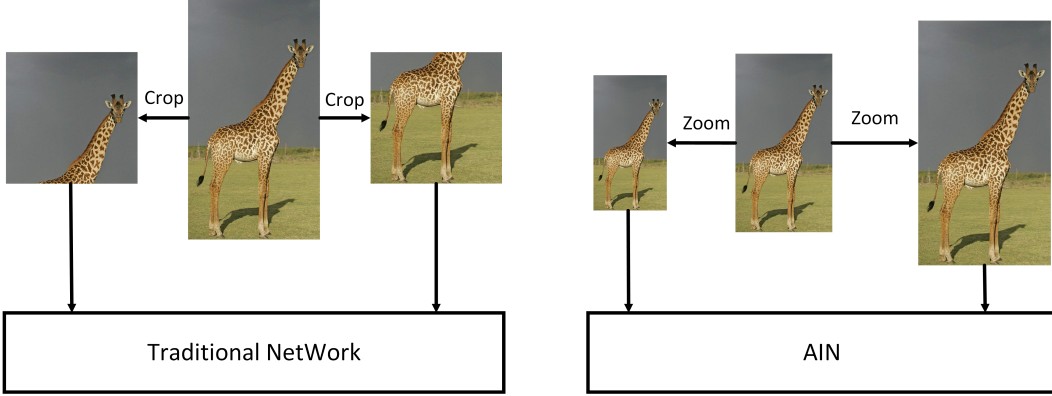

Figure 7: Comparision between crop and zoom.

## 4.1 CIFAR10

The CIFAR-10 datasets consist of 60,000 with fixed color images size of 32×32. 10 object classes with 50,000 training images and 10,000 test images. The broadly applied state-of-the-art network structure DenseNet is used as baseline method. To make a fair comparison, we name the original dataset as "C10" without any data argumentation. Then we adopt a standard data augmentation scheme (mirroring/shifting), and keep most of the setting the same as in ResNet Paper (He et al., 2016) and DenseNet Paper (Huang et al., 2016). We denote this data augmentation scheme by a + mark at the end of the dataset name as "C10+". We didn't try the experiment of "C10*" for the reason that images in CIFAR-10 have the constant size.

We train all these models using *SGD* optimizer with a mini-batch size of 64, set the initial learning rate to 0.1. The learning rate is divided by 10 at 150epoch, 225epoch. We terminate training at 300epoch. The main results on CIFAR10 are shown in Table. 2. To highlight general trends, we mark all results that outperform the baseline in **boldface** and the best result in **blue**.

## 4.2 FURNITURE-128.

The training dataset includes images from 128 furniture and home goods classes. It includes a total of 194,828 images with 26652 various image size for training (e.g. 350×350, 640×480, 268×400) and 6,400 images with 1606 various image size for validation and 12,800 images for testing. We do the image preprocess in two ways: The first way is to wrap the image into a fixed size 224×224

Table 2: Error rates (%) on CIFAR

| Motheds | Depth | Params | C10 | C10+ |
|---|---|---|---|---|
| AlexNet | 8 | 4.5M | 35.8 | 26.4 |
| ResNet-110 | 110 | 1.7M | - | 6.43 |
| DenseNet(k=12) | 40 | 1.04M | 7.00 | 5.36 |
| DenseNet(k=12) | 100 | 7.08M | 5.84 | **4.21** |
| AIN(CNNBlock) | 10 | 4.7M | 21.8 | 14.2 |
| AIN(ResBlock) | 110 | 2.31M | - | 6.26 |
| AIN(DenseBlock) | 40 | 1.57M | 6.72 | 5.15 |
| AIN(DenseBlock) | 100 | 10.6M | 5.61 | **4.02** |

associated with the standard data augmentation scheme (mirroring/shifting) as (He et al., 2016; Huang et al., 2016), we name this dataset as "furniture128+". The second way is to scale the image into the maximum slide size of 224(e.g.:350×350 → 224×224, 640×480 → 224×168, 268×400 → 150×224). With random mirroring and zoom fact between 0.7 and 1.3. The input data still remain the size distinctive, we name this dataset as "furniture128*".

We train all these models using *adam* optimizer with a mini-batch size of 256, set the initial learning rate to 0.005. The learning rate is divided by 10 at 38epoch, 53epoch. We terminate training at 80epoch. For DenseNet and Resnet, used the pre-trained model for initialization, with the input image scale into the fixed size. "furniture128*" can only be trained in AIN cause of the various of image size. The main results on "Kaggle-furniture" are shown in Table. 3. The comparison between AIN and the base-line model in loss and error are shown in Fig. 8 and Fig. 9.

Table 3: Error rates (%) on furniture-128

| Motheds | Depth | Params | furniture-128+ | furniture128* |
|---|---|---|---|---|
| ResNet-101 | 101 | 42.9M | 20.1 | - |
| ResNet-152 | 152 | 58.6M | **18.8** | - |
| DenseNet121 | 121 | 7.2M | 18.1 | - |
| DenseNet169 | 169 | 12.9M | **16.4** | - |
| AIN(ResBlock) | 101 | 44.1M | 19.3 | 18.7 |
| AIN(ResBlock) | 152 | 59.9M | 18.2 | 18.0 |
| AIN(DenseBlock) | 121 | 7.56M | 17.1 | 16.5 |
| AIN(DenseBlock) | 169 | 14.8M | 16.2 | **15.3** |

Figure 8: Furniture-128 loss and epochs

Figure 9: Furniture-128 error and parameters

### 4.3 IMAGENET

The ILSVRC 2012 classification dataset (Deng et al., 2009) consists 1.2 million images with 102,286 various size for for training (top-3 size :500×375, 375×500, 500×333), and 50,000 images with 5212 various size for validation from 1, 000 classes. We train ImageNet+ by DenseNet with random crop and single crop and 10-crop in validation set the same as (Huang et al., 2016). We train ImageNet* by AIN With zoom fact between 0.8 and 1.2 for in training set, without zoom and 10-zoom while testing, we add a dropout layer after GAIL and set the dropout rate to 0.2 to prevent overfitting.

We train models for 90 epochs with a batch size of 256. The learning rate is set to 0.1 initially, and is lowered by 10 times at epoch 30 and 60. Results are show in Table. 4. AIN121(AIN169, AIN201) refers that the setting of convolutional blocks are exactly the same as DenseNet121(DenseNet169, DenseNet201).

Table 4: The top-1 and top-5 error rates(%) on the ImageNet validation set, with single-crop / 10-crop testing for ImageNet+, without zoom / 10- zoom testing ImageNet*

| Motheds | Params | top1 | top5 |
|---|---|---|---|
| DenseNet121(ImageNet+) | 8.06M | 25.02/23.61 | 7.71/6.66 |
| DenseNet169(ImageNet+) | 14.31M | 23.80/22.08 | 6.85/5.92 |
| DenseNet201(ImageNet+) | 20.24M | 22.58/**21.46** | 6.34/**5.54** |
| AIN121(ImageNet*) | 8.20M | 23.77/23.23 | 6.73/6.61 |
| AIN169(ImageNet*) | 15.99M | 22.24/21.99 | 6.06/5.90 |
| AIN201(ImageNet*) | 22.48M | 21.45/**21.23** | 5.54/**5.43** |

It's obvious to find that 10-crop achieve a lower error rate than single-crop in previous previous literature such as (Huang et al., 2016; Chollet, 2016; He et al., 2016). Table. 4 shows that the gap between without-zoom/10-zoom is much smaller than the gap of single-crop/10-crop. For DenseNet201 top1 error, single-crop give an error rate of 22.58 ,and 21.46 for 10-crop. There is a gap of 1.12 between them. The top1 error gap between without-zoom and 10-zoom of AIN201 is 0.22 which is much smaller than 1.12. Crop for 10 times almost certainly get the global vision of the image statistically which increase the 10 times computational cost in forward pass. AIN can get the global vision of the image with single forward pass, as a result the improvement of 10-zoom is not so significant than 10-crop. It infers that random crop might lost some information of the image, and AIN is a better way to deal with the various image size comparing with random crop.

## 5 DISCUSS

In this work, we introduce AIN which is a flexible solution for handling different scales, sizes, and aspect ratios used attention mechanism. By visualizing the weight matrix in the LAIL (Fig. 6), we can know that the network have learnt the key information of the object. Thus it's not surprise to get a better result in CIFAR10 and Kaggle-*Funiture128* and ImageNet than the same model using crop/scale dealing with the image size problem.

We believed that AIN can reach a higher accuracy by more detailed tuning of hyper-parameters and learning rate schedules. With the property of dealing with various image size, AIN have the potential for other computer vision task especially object detection problem. There is still room for improvement in AIN. We are doing continuous improvement in our network and expect it to have a perfect performance in the future work.

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
