# OpenReview forum: "ATTENTION INCORPORATE NETWORK: A NETWORK CAN ADAPT VARIOUS DATA SIZE"
_ICLR.cc/2019/Conference_

### Official Review · AnonReviewer1 · 2018-10-26
**Hard to understand and weak methodological contribution**

**Rating:** 2
**Confidence:** 4

**Review:**

General comment
==============
The authors describe an attention mechanism for training with images of different sizes. The paper is hard to understand due to major grammatical errors and unclear descriptions. Methods for training with images of different sizes have been proposed before, e.g. spatial pyramid networks. I also have concerns about their evaluation. Overall, I believe that the paper is not ready to be submitted to a conference or journal.

Major comments
=============
1. Methods for training with images already exists, e.g. spatial pyramid pooling (http://arxiv.org/abs/1406.4729) or fully-convolutional networks (https://people.eecs.berkeley.edu/~jonlong/long_shelhamer_fcn.pdf). These are not cited in the paper and not included as baselines in their evaluation.

2. The attention mechanisms looks similar to classificat soft-attention (https://arxiv.org/abs/1502.), which is not cited in the paper.

3. The paper contains major spelling and grammatical errors, making it hard to understand important aspects.

4. I can not see a clear improvement of their method over ResNet and DenseNet when the same number of model parameters is about the same. Without making sure that the number of model parameters is about the same, it is unclear if the performance gain is due the increased number of model parameters or the methodology.

---

### Official Review · AnonReviewer3 · 2018-10-31
**Another architecture variant of ConvNets without sufficient experimental benchmark**

**Rating:** 4
**Confidence:** 4

**Review:**

I was excited about the title and abstract but my expectation started to fall as I parsed through the main text. Here are some of my major concerns:

1. The entire text is plagued by syntax errors that sometimes inhibit the narrative and prevent the proper understanding.

2. Section 3 explains the architecture clearly, but fails to justify, perhaps in theory or at least in intuition, why AIN could have any advantage with such a distinct choice of parameterization. Nor can I find any solid evidence from the experiments that this is indeed the case.

3. Section 4 seems ad-hoc, simply presenting tables without ablation study makes it hard to trust the proposed architecture.

---

### Official Review · AnonReviewer2 · 2018-11-03
**A method to deal with the problem of fixed input image sizes in CNNs classifiers**

**Rating:** 3
**Confidence:** 5

**Review:**

This paper presents a strategy to overcome the limitation of fixed input image sizes in CNN classifiers. To this end, the authors incorporate some local and local attention modules, which fit inputs of arbitrary size to the fixed-size fully connected layer of a  CNN. The method is evaluated on three public classification benchmarks: CIFAR-10, ImageNet and Kaggle-Furniture128.

The results are better than those of the baseline architecture with fixed input size.

Even though the need of handling arbitrary input size is an interesting problem, I have several major concerns about this paper:

- One of the main problems of this paper is its presentation, both the writing and methodology. The writing is very poor, with continuous errors and many wrong definitions and concepts. For example, authors talk about ‘data argumentation’, ‘pooling reduces the size of the hidden layers’,’back-to-back convolutional layers’

Further, the paper is not well structured, which makes it very hard to follow.

Methodology:

Another major concern is that I do not see how this approach allows the network to be input-size independent. If one looks at table 1, in both AIN-121 and AIN-169 the GAIL module employs kernel sizes equal to M/32xN/32, with M and N denoting the input image sizes. In this case, for each image, the kernel size will be different and, consequently, the number of learnable parameters. It is not clear to me how this is solved in this paper, as it ultimately results in a ‘different’ architecture for each different input size.

When doing the sum on the proposed module, what does the result represent? absolute sum? mean of the sum? I also believe that a lot of information is lost when performing this operation (for example going from 32 to 1), in addition of the other spatial reductions during the network forward pass. Please comment on this and give a more detailed information about the proposed module.

Evaluation:
In CIFAR-10, authors say that ‘keep MOST of the setting similar to ResNet’. What is then difference with the training with ResNet? For a fair comparison both settings should remain the same. In addition, what is the benefit of evaluating this approach on CIFAR-10, as the images are all of the same size? Furthermore, improvement is marginal with respect to the baselines (and it is not clear what is the reason behind the improvement), while increasing the model complexity by nearly 50%.

Kaggle-Furniture128: Why the learning is stopped exactly at epochs 38 and 53? Is this the same for all the networks? DenseNet and ResNet are pre-trained with what dataset?

ImageNet: In table 4, while the results for the baselines are evaluated on the validation set, the test set is used for evaluating the proposed approach. Furthermore, some results on the test set are obtained with ‘augmentations’. The reported values should correspond to the original test set without any kind of modification.

Minor comments:

The authors assess the input fixed-size problem as a main problem in image processing. Despite being a limitation, some other image processing tasks, such as semantic segmentation, do not suffer from this problem, as CNNs are fully convolutional, and can accommodate images of arbitrary size.

Many inconsistencies between terms: LAIL and then LAIN and GAIL and GAIN.'

---

### Meta-Review · Area_Chair1 · 2018-12-10
**metareview: no rebuttal**

**Confidence:** 5
**Recommendation:** Reject

**Metareview:**

All reviewers agree that the paper should be rejected and there is no rebuttal.